# Transcriptional Regulation of *RUNX1*: An Informatics Analysis

**DOI:** 10.3390/genes12081175

**Published:** 2021-07-29

**Authors:** Amarni L. Thomas, Judith Marsman, Jisha Antony, William Schierding, Justin M. O’Sullivan, Julia A. Horsfield

**Affiliations:** 1Department of Pathology, Dunedin School of Medicine, University of Otago, Dunedin 9016, New Zealand; amarni.thomas@otago.ac.nz (A.L.T.); jisha.antony@otago.ac.nz (J.A.); 2Department of Cardiology, University Medical Centre Utrecht, 3584 CX Utrecht, The Netherlands; J.Marsman@umcutrecht.nl; 3The Maurice Wilkins Centre for Biodiscovery, The University of Auckland, Private Bag 92019, Auckland 1142, New Zealand; justin.osullivan@auckland.ac.nz; 4Liggins Institute, The University of Auckland, Private Bag 92019, Auckland 1142, New Zealand; w.schierding@auckland.ac.nz; 5MRC Lifecourse Epidemiology Unit, University of Southampton, Southampton SO17 1BJ, UK; 6Genetics Otago Research Centre, University of Otago, Dunedin 9054, New Zealand

**Keywords:** acute myeloid leukaemia, *RUNX1*, transcription, enhancer, silencer, chromatin

## Abstract

The *RUNX1/AML1* gene encodes a developmental transcription factor that is an important regulator of haematopoiesis in vertebrates. Genetic disruptions to the *RUNX1* gene are frequently associated with acute myeloid leukaemia. Gene regulatory elements (REs), such as enhancers located in non-coding DNA, are likely to be important for *Runx1* transcription. Non-coding elements that modulate *Runx1* expression have been investigated over several decades, but how and when these REs function remains poorly understood. Here we used bioinformatic methods and functional data to characterise the regulatory landscape of vertebrate *Runx1*. We identified REs that are conserved between human and mouse, many of which produce enhancer RNAs in diverse tissues. Genome-wide association studies detected single nucleotide polymorphisms in REs, some of which correlate with gene expression quantitative trait loci in tissues in which the RE is active. Our analyses also suggest that REs can be variant in haematological malignancies. In summary, our analysis identifies features of the *RUNX1* regulatory landscape that are likely to be important for the regulation of this gene in normal and malignant haematopoiesis.

## 1. Introduction

Accurate spatiotemporal and quantitative gene expression is crucial for normal development and, in many cases, is achieved by the interaction of promoters with *cis*-regulatory elements (REs). REs, such as enhancers, are able to control the expression of genes by long-range chromatin interactions [1,2].

Most REs evolve rapidly and are rarely conserved at the DNA sequence level among species due to positive evolutionary selection [3]. However, clusters of conserved REs surround some highly conserved developmental genes. REs can be highly tissue-specific, and surprisingly remote from their gene targets [4,5]. Although REs often regulate the closest gene, they can also control expression of genes further afield [6,7,8]. Fewer than 50% of enhancers contact the nearest gene promoter [9]. Long-range chromatin interactions between promoters and REs can be mediated by scaffolding proteins and transcription factors (TFs) to regulate gene expression [10]. These factors include those responsible for the three-dimensional organisation of chromatin, such as cohesin and CTCF [10,11].

The transcription factor Runx1 is crucial for definitive haematopoiesis [12,13]. In humans, the *RUNX1* gene is frequently targeted by translocation or mutation in acute myeloid leukaemia (AML) and other myeloproliferative disorders [14,15,16]. Expression of the *Runx1* gene involves two alternative promoters, P1 (distal) and P2 (proximal) (Figure 1). These two promoters are differentially regulated to produce alternative transcripts that are translated into different protein products that function in neuronal as well as haematopoietic development ([17], reviewed in [18]). 

Both haematopoietic and non-haematopoietic expression of *Runx1* appears to rely on REs. For example, the *Runx1* promoters by themselves cannot drive haematopoietic expression of *Runx1* [19]. Rather, a well-characterised enhancer located ~24 kilobases (kb) downstream of the transcription start site of *Runx1*’s P1 promoter regulates *Runx1* expression specifically in haematopoietic cells. In mouse, this RE is variously termed +23/+24/eR1/RE1 [19,20,21,22,23], and is responsible for the activation of *Runx1* expression in haematopoietic stem/progenitor cells. *Runx1* +23/+24 also acts as a haemogenic endothelial cell-specific enhancer in mouse and zebrafish embryos and is therefore is highly conserved in eukaryotes [21,23,24,25]. Bee et al. (2009) showed the +23/+24 enhancer works with both promoters to drive haematopoietic stem cell-specific gene expression [19]. Interestingly, +23/+24 was also described as a silencer in HEK293 cells [23].

Other putative REs were identified upstream of *Runx1*, and between the P1 and P2 promoters in humans and mice (Figure 1) [21,23,24,25,26,27,28,29,30,31,32,33]. Many of these REs appear to be important for normal haematopoiesis, and dysregulated in leukaemia. Cheng et al. (2018) found that disruption of a RE by chromosomal translocation can upregulate *RUNX1* and progress leukaemogenesis [28]. Mill et al. (2019) deleted the whole intron between *RUNX1* P1 and P2 in OCI-AML5 cells [34], removing three described REs including +23/+24. The majority of the edited cells were eradicated via apoptosis, and the viable edited cells had significantly decreased *RUNX1* expression and slower growth [34].

Recent studies have identified recurrent mutations in REs of genes associated with oncogenesis, such in as the *TAL1*, *ETV1*, and *PAX5* enhancers [35,36,37]. In AML and other myeloid malignancies where no mutation is found in the coding sequence, there may instead be mutations in REs that affect gene function, including *RUNX1* [38]. Therefore, defining the compendium of conserved *Runx1* REs is imperative to understanding the regulatory landscape of *Runx1* in normal and malignant haematopoiesis.

Here we used bioinformatic methods and functional data to identify possible REs that are conserved between human and mouse. Our analyses also suggest that *RUNX1* REs are indeed affected by mutation in haematological malignancies.

## 2. Materials and Methods

### 2.1. RE sequence Identification 

Genomic sequences surrounding the *Runx1* locus for human (*Homo sapiens*) and mouse (*Mus musculus*) were obtained from publicly available genome assemblies [human (GRCh38/hg38 and GRCh37/hg19) assembly, and mouse (mm9 and mm10) assembly] on the UCSC genome browser [39,40]. Sequences were determined based on primer information or chromosome coordinates provided in the identifying paper [21,23,24,25,26,27,28,29,30,31,32,33].

### 2.2. Epigenetic Analyses Using ENCODE Data

ENCODE data were used to detect the presence of various histone modifications, DNase I hypersensitivity sites, TFs and cohesin binding sites [27,41,42,43]. Analysis of ENCODE annotations was carried out using data submitted by Stanford, Yale, University of Washington (UW) and Ludwig Institute for Cancer Research (LICR). To determine the RE locations in different genome assemblies the UCSC ‘liftOver’ tool was used to convert coordinates within a species [42].To determine the human genome co-ordinates for identified *Runx1* enhancers, the sequences were searched using the UCSC BLAT and NCBI BLAST tools [44,45]. Further genomic features of the conserved REs were deduced by analysing the human FANTOM 5 data [46,47] in the UCSC Genome Browser.

Chromatin State predictions (ChromHMM) for K562 cells were used to annotate the REs based on epigenetic information [48]. Further regulatory markers for K562 cells were uploaded into the UCSC genome browser including; super-enhancers [49], silencers [50], cohesin mediated chromatin accessibility [51].

### 2.3. Single Nucleotide Polymorphism (SNP) Analysis

To assess the phenotypic and functional relevance of any reported SNPs in each region, the GWAS catalog (GRCh38/hg38) [52], Catalogue of somatic mutations in cancer (COSMIC) GRCh37/hg19 and GRCh38/hg38) [53], DICE (GRCh37/hg19) [54,55] and HaploReg v4.1 (GRCh37/hg19) [56] were used. The GWAS catalog highlights any publications that report on the association of that SNP with a phenotype. The HaploReg v4.1 tool displays the ChromHMM, genome characteristics, and TF binding data for each SNP and predicts regulatory motif changes. To establish if any SNPs had an association with changes of gene expression (expression Quantitative Trait Loci–eQTL), we searched all SNPs with a minor allele frequency (MAF) of >1% in the Genotype-Tissue Expression (GTEx) portal v8 (GRCh38/hg38) (https://gtexportal.org/home/ accessed on 25 June 2021). The gnomAD browser [57] v2.1.1 (GRCh37/hg19) and v3 (GRCh38/hg38) reviewed SNPs with MAF >1% for any ClinVar associations and/or publications. To identify physical interactions between SNPs in the REs and genes, the CoDeS3D pipeline [58] was used to interrogate Hi-C chromatin interaction libraries. 

### 2.4. Comparative Analysis of Cancer Genomic Datasets

To assess whether AML patients had mutations in the conserved REs, publicly available cancer genomic datasets were explored. Online tools were used to categorise publicly available patient information into different tumour types for analysis. The large-scale cancer genomics datasets were retrieved from International Cancer Genome Consortium (ICGC) Data Portal [human Feb. 2009 (GRCh37/hg19) assembly] (Whole Genome sequence (WGS) data for 1732 AML/MDS donors) [59,60], The c-BioPortal for Cancer Genomics [human Feb. 2009 (GRCh37/hg19) assembly] (50 WGS/451 total AML samples) [61,62], COSMIC database (GRCh37/hg19 and GRCh38/hg38) (39 additional WGS AML samples) [53] and The Cancer Genome Atlas (TCGA) Data Portal [human Mar. 2006 (NCBI36/hg18) assembly] (50 WGS AML samples) [63].

### 2.5. Prediction the Functional Consequences of Non-Coding Variations

To predict the pathogenicity of SNPs and cancer associated genetic variations within R1REs Functional Analysis through Hidden Markov Models (FATHMM v2.3) (GRCh37/hg19) was used [64,65].

## 3. Results

### 3.1. Conserved Human RUNX1 Regulators

We first sought to identify enhancers for *Runx1* that are conserved between human and mouse. In mice, previously studies identified 29 possible regulators of *Runx1* (−371, −368, −354, −328, −327, −322, −321, −303, −181, −171, −101, −59, −58, −48, −43, −42, +1, +3, +24, +32, +59, +64, +87, +99, +110, +171, +181, +199, +204) [21,24,25,31,32,33,66]. Several studies have focused on a strong *Runx1* enhancer in mice: *Runx1* +23/+24/eR1/RE1 [19,20,21]. The naming of the element refers to the number of kilobases (kb) downstream of a reference point. The inconsistency of enhancer naming between groups is because Ng et al. (2010) uses the transcriptional start site (TSS), +1 as the reference point [21], whilst the other groups refer to this enhancer being 23.5 kb downstream from the ATG of exon 1. +23/+24 is also referred to as eR1 (enhancer of *Runx1* [22]), and RE1 (regulatory element 1 [67]). Hereafter, +23/+24/eR1/RE1 will be referred to as *Runx1*
regulatory element 1 (R1RE1). 

Markova et al. (2011), Cauchy et al. (2015), Gunnell et al. (2016) and Cheng et al. (2018) and Vukadin et al. (2020) identified 9 potential human *RUNX1*
regulatory elements (R1REs) (−250, −188, −139, −57, +24, +43, +62, +139, intron 5.2). Some of these discoveries included functional characterisation and identification of long-range interactions with *RUNX1* promoters [23,26,28,29,30].

When gathering sequence information about these mouse REs we noted that three regions identified by Schütte et al. (2016) are the same as regions described by Marsman et al. (2017): Schütte’s −328 is Marsman’s −327; Schütte’s −322 is Marsman’s −321; and Schütte’s −59 is Marsman’s −58. The *Runx1* m+204 sequence sits within the previously described *RUNX1* intron 5.2.

Based on sequence analyses we determined that 12 out of the 29 previously identified mouse *Runx1* REs are conserved in human. This increases the number of ‘known’ putative human R1REs from 9 to 21. The *Runx1* enhancers that were identified to be conserved between human and mouse were assigned a common identity, R1RE2–R1RE21, and (other than R1RE1) they are numbered in order of 5’ to 3’ location relative to the orientation of the *Runx1* gene (Table 1). 

Conservation between mice and humans suggests that these REs are fundamental for the correct regulation of *RUNX1*. To determine possible regulatory function in human haematopoietic cells, bioinformatic analysis of R1REs in K562 chronic myelogenous leukaemia cells was undertaken (Table 2, Figure 2). K562 cells express *RUNX1* and are comprehensively annotated in ENCODE. Our analysis showed that not all R1REs appear to be active in K562 cells. Moreover, the accessibility of some REs was altered upon cohesin mutation, indicating that cohesin’s role in 3D genome structure might influence the function of some REs. 

### 3.2. Chromatin Features of R1REs

ChromHMM indicates that R1RE4 and R1RE5 are in repressed chromatin, and that R1REs 1–3, 6, 8, 10–12, and 14–18 are active enhancers. R1REs 9, 13, 19 and 21 show chromatin features of enhancers; however, they are labelled as transcription-associated alongside R1REs 7, 8 and 20 (Table 2, Figure 2). Transcription association may be due to R1RE proximity with exons when compared to the regions annotated as enhancers or it may indicate that these regions are producing RNA, for instance enhancer RNA (eRNA). 

R1RE1 has strong enhancer characteristics and recruits haematopoietic TFs SPI1, TAL1, GATA1 and GATA2. SPI1 also located to R1REs 7, 8 and 19. R1RE19 also recruits TAL1 in haematopoietic progenitors. Haematopoietic TFs TAL1, GATA1 and GATA2 bind to R1REs 2, 10 and 18. These regions showed TF binding sites and chromatin modifications similar to that found at R1RE1 (Table 1, Figure 2).

R1RE1 also recruits LSD1, a mediator of transcriptional repression usually seen in silencers. Interestingly, 17 other REs (R1REs 2, 3, 5, 6, and 9–21) had LSD1 binding in K562 cells. Cheng et al. (2018) inhibited LSD1 in three haematopoietic cell lines (K562, OCI-AML3 and U937) and observed upregulation of *RUNX1,* implying that LSD1 normally represses *RUNX1* [28]. Kerenyi et al. (2013) previously showed Lsd1 represses key haematopoietic stem cell genes including *Runx1* [69]. 

Mouse ATAC-seq [66] highlighted altered accessibility at R1REs in different cell types. R1REs 1, 4, 6, 10, 13 and 18 were accessible in mesoderm cells; whereas R1REs 1, 10, 13, 18, and 21 were accessible in haematopoietic progenitor cells (HPC). These data reflect differential tissue-specific activity of R1REs.

ATAC-seq in K562 cells with a null mutation in the *STAG2* gene, which encodes a cohesin subunit [51], showed that R1REs 2, 6, 18 and 21 regions reside in differentially open chromatin regions in *STAG2* mutant cells (Table 2). This suggests that cohesin influences the accessibility (and possibly the function) of these REs. R1RE1 binds cohesin subunit SMC3 and CTCF, while cohesin subunit RAD21 binds with CTCF at R1RE12, and in the absence of CTCF with R1RE18. CTCF binding was also observed in R1REs 2, 10–11, 13, 15–17, and 19–21 (Table 2). 

### 3.3. Enhancer RNA (eRNA)

Enhancer RNA (eRNA) has been shown to be involved in functional roles including; assisting enhancer-promoter looping formation, supporting pause-release of RNA Pol II which facilitates transcription elongation, promoting TF and co-regulator binding and aiding target gene transcription [70,71,72,73,74,75,76,77,78,79,80]. Although eRNA is typically used for enhancer identification, not all active enhancers produce detectable eRNA [81,82].

FANTOM5 data showed that 16 of the 21 R1REs produced eRNA in haematopoietic cells (Figure 2, Appendix A), whereas R1REs 3, 6, 7, 9 and 10 do not have any reported eRNA expression. ChromHMM labelled R1REs 7–9, 13, and 19–21 as transcription-associated. This may be attributed to enhancer RNA transcription, as mR1REs 8, 13, and 19–21 all produce eRNAs. It remains to be determined whether the transcription association observed in R1REs 7 and 9 is also due to enhancer RNA production.

### 3.4. Single Nucleotide Polymorphism (SNP) Analysis of R1REs

Disease-associated variants are found in REs and in genes with equal frequency [83]. More than 95% of genome-wide association studies (GWAS) single nucleotide polymorphisms (SNPs) are located in intergenic regions, of which over 75% are associated with open chromatin (DNase I HS sites) implying a strong association with REs [84]. A variant in a RE may cause differential regulation of a gene, and could be functionally equivalent to a mutation in the coding sequence of the gene itself. Recent studies have shown that alterations to REs are associated with dysregulation of oncogenic genes [85]. It is therefore possible that mutations in enhancers or insulators may change their function, and subsequently could lead to altered gene expression [86].

Single nucleotide polymorphisms (SNPs) are present in each of the human *RUNX1* conserved REs, except R1REs 1, 9, 12, 16, 17 and 20 (Figure 2, Appendix A). SNPs were reported in R1REs 2, 3, 10, 14, 18 and 21 in haematopoietic, brain, thyroid, skin and oesophagus cells. 

Of the 40 SNPs identified in the conserved R1REs, 6 had previously reported functional effects in haematopoietic cells. R1RE2 contains rs2834945, a T to C change that is predicted to affect 11 regulatory motifs including GATA2. rs2834945 affects the expression of kynurenine 3-monooxygenase (KMO) in peripheral blood monocytes [87]. KMO encodes a mitochondrial outer membrane protein that catalyses the hydroxylation of L-tryptophan metabolite, L-kynurenine, to form L-3-hydroxykynurenine [88]. High KMO expression leads to neurodegeneration [89]. 

R1RE3 harbours two functional SNPs: rs16993221 and rs909143. rs16993221 is associated with white blood cell count and contributes to chronic inflammation [90]. The A to T change in rs16993221 alters two regulatory motifs, BATF and IRF, which work together in immune response. The A to G change in rs909143 is also related to expression of KMO in peripheral blood monocytes [87], and is predicted to affect two regulatory motifs (NR3C1, POU2F2). R1RE18 harbours rs73900579, a T to C variant that alters two regulatory motifs and is associated with red cell distribution width [91]. R1RE21 contains two functional SNPS, rs2249650 and rs2268276; both are associated with AML susceptibility. The different SNPs are in linkage disequilibrium (LD) and change the ability of R1RE21 to act as an enhancer. A (rs2249650) G (rs2268276) bases and AA have greater enhancer capability than GA and GG. GA has the least enhancer capability. The SNPs also affect SPI1 binding capability; AG has strong SPI1 binding, GA has weak SPI1 binding, whereas AA and GG have medium SPI1 binding capability [92].

### 3.5. Cancer Associated Genetic Variation Occurrence in R1REs

Although not all SNPs found in this study have been functionally analysed, some of the R1RE SNPs were reported in patients with AML. Two AML patients had R1RE5 SNP rs2834885. One of the patients who had R1RE5 SNP rs2834885 also had three additional alterations to R1REs; R1RE2, R1RE14 SNP rs9976900 and R1RE19 SNP rs2284613. R1RE14 SNP rs9976900 has previously been associated with long non-coding RNA *RUNX1-IT1* eQTL in the brain cortex [93] and was also associated with paediatric asthma [94].

Two SNPs in R1RE15 (rs933131 and rs2834716) were also found in patients with AML. Interestingly, the patient with R1RE15 rs2834716 also had the R1RE21 SNPs (rs2268276 and rs2249650) which were previously associated with AML susceptibility [92]. Five patients with malignant lymphoma had mutations within R1REs 1 and 10 that are not reported as SNPs with a minor allele frequency (MAF) of >1%. Three variants in three different patients were found in R1RE18, however, these variants are not reported as SNPs with a MAF of >1%. Four patients had R1RE10 variants chr21:g.36480715C>A (rs199811665), chr21:g.36480761A>C and chr21:g.36481470C>T (rs1440069314). One case of malignant myeloma reported a variant within R1RE1, chr21:g.36399191A>T. Variants were also found in R1RE15 (chr21:g36359341T>G), R1RE18 (chr21:g36280880A>G) and R1RE21 (chr21:g36180724G>A).

The finding that SNPs and alterations in R1REs associates with human disease raises the possibility that these regulatory elements may be important for regulation of *RUNX1* and/or other genes. It is currently unclear whether these variants have a higher or lower frequency than if they occurred by chance, and a more comprehensive catalogue of genomic data would be needed to provide significance. R1RE21 SNPs were found to be associated with AML susceptibility. However, it is possible that mutations in R1REs may explain AML progression in patients who have no known protein-coding mutations. In support of this idea, data available at cBioportal [61,62] shows a wide variability in *RUNX1* expression, regardless of *RUNX1* mutation status (Appendix A).

Functional analyses are necessary to determine whether sequence variation is causative of differential gene expression and consequently, disease or neoplasia.

### 3.6. Prediction the Functional Consequences of Non-Coding Variations 

The majority of the R1REs cancer associated genetic variations and SNPs identified in these regions have no functional data. Predictions of the functional consequences of R1REs cancer associated variations and SNPs highlight 7 pathogenic predicted variations (Appendix A). R1RE1 variation (chr21:g.36399191A>T), R1RE3 (rs116951441), R1RE11 (rs2834756), R1RE18 (rs73900579), R1RE19 (rs2284613) and the previously AML susceptibility associated SNPs R1RE21 (rs2249650 and rs2268276) all have prediction scores of >0.87. While the FATHMM algorithm assigned pathogenicity to previously uncharacterised variants, it did not predict pathogenicity for SNPs in R1RE3, R1RE10, and R1RE14 that were associated with eQTLS or other phenotypes. This highlights the importance of using several approaches to determine the effect of variants in R1REs. Importantly, FATHMM predictions reinforced the pathogenic potential of SNPs in R1RE18 and the leukaemia-associated SNPs in R1RE21. The combined analyses show that the R1RE3, R1RE18 and R1RE21 are affected by changes that have potential to be detrimental to correct gene regulation. 

## 4. Discussion

In this study, we identified 12 *Runx1* regulatory regions that are conserved between human and mouse, and are likely to be important for controlling expression of human *RUNX1*. This analysis increased the defined total of *RUNX1* enhancers in human from 9 to 21, compared with 29 known REs in mice (Figure 1 and Figure 2 and Table 1). Previously, both mouse and human *Runx1* REs were named according to their distance from the TSS (or ATG) of the P1 promoter of *Runx1*. Here we instead assign each of the conserved REs a number, starting with R1RE1 through R1RE21. R1RE1 was assigned to +23/+24/eR1/RE1 because it is the best characterised of the REs, with other REs named in order according to their 5’-3’ location on the same strand as the *Runx1* gene. The assignation of a common identifier for enhancers conserved between human and mouse may eliminate confusion arising from different nomenclature in the numerous studies that seek to identify and characterise *RUNX1* enhancers.

Of the identified R1REs, 8 interact with the *RUNX1* promoters; R1REs 1, 2, 4, 10, 15, 18 and 21 have differing interactions with *Runx1* P1, P2 and R1RE1 in different mouse or human cell lines (Table 1) [23,25,27,28]. A recent preprint describes a study in which enhancer-promoter contacts at the mouse *Runx1* gene were determined step-wise during differentiation of mouse embryonic stem cells (ESC) to mesoderm and then on to HPC [66]. During mesoderm specification, the P2 promoter increased interactions with R1REs 1, 4, 6, 13, and 18 (and additional mouse enhancers not conserved in human). Upon differentiation of mesoderm to HPC, contacts were lost with the more upstream R1REs, while contacts to both P1 and P2 were increased with R1REs 11 and 21. Concomitantly, the *Runx1* gene was expressed from the P2 promoter upon differentiation to mesoderm, and both P1 and P2 upon adopting HPC identity [66]. This study therefore reinforces the assumption that conserved REs for *Runx1* are important for expression of *Runx1* during haematopoietic differentiation. 

The regulatory activity of 15 of the 21 human R1REs has been functionally confirmed. Moreover, there is good evidence that R1REs 16 and 21 are directly involved in AML disease progression. Cheng et al. (2018) characterised R1RE15 as a silencer of *RUNX1* P2 by measuring the repressive effect of various deletion/mutant R1RE15 luciferase constructs on transcription from the P2 promoter in K562, OCI-AML3 and U937 cells [28]. However, in K562 cells R1RE15 has ENCODE and ChromHMM annotations associated with enhancer activity. Regions that have both enhancer and silencer annotations may have multiple roles depending on DNA interaction partners and the cell type. The discovery of *RUNX1* silencer R1RE15 resulted from characterisation of a novel t(5;21)(q13;q22) translocation involving *RUNX1* that was acquired during the progression of myelodysplastic syndrome to AML in a paediatric patient [28]. This translocation did not generate *RUNX1* fusion, but rather it aberrantly upregulated the P2 isoform of *RUNX1*. The authors state that the translocation facilitated upregulation of *RUNX1* P2 because the silencer at R1RE15 was removed. 

R1RE21 contains SNPs that are associated with AML susceptibility, and in addition, R1RE21 is the site of translocation in t(8;21) in AML patients in which *RUNX1* and *ETO* genes recombine. Schnake et al. (2019) identified a long non-coding RNA in R1RE21, at a site of these frequent chromosomal translocations [95]. This long non-coding RNA results in a more relaxed chromatin organisation at R1RE21 (at the location of the breakpoint). Consequently, it is possible that chromatin relaxation leads to a higher probability of double stranded breakages in myeloid cells, resulting in translocations [95].

There were no SNPS with MAF of >1% present in R1RE1. The one detected mutation in R1RE1 (chr21:g.36399191A>T) is predicted by FATHMM to have a pathogenic function. This suggests that there is strong selection pressure to conserve R1RE1. Consistent with this, the region containing R1RE1 appears to be functionally important in haematopoietic cells. To determine the importance of R1RE1 for *RUNX1* expression and function, Mill et al. (2019) deleted the P1-P2 intron of *RUNX1* P1 in OCI-AML5 cells [34], which led to a strong selection against edited cells. The P1-P2 intronic region, which contains multiple R1REs, is therefore important for cell survival and *RUNX1* expression.

Predictions of functional consequences of genetic variation within the R1REs estimate that 7 variations are likely to be pathogenic. Further functional analysis of the genetic variations will allow researchers to determine the roles of these R1REs in normal haematopoiesis as well as disease progression. These predictions may be an underestimation of the functional consequences of genetic variation, as the they are calculated based on available ENCODE data.

The ability to identify significant SNPs and mutations in haematopoietic cells is limited by the scarcity of whole genome sequence in individual haematopoietic cell types, AML, and other myeloid malignancy samples. Therefore, SNP frequency in R1REs could be an underestimate. The present study is limited by both the paucity of genomic sequence data of REs in normal and leukaemic cells, and the lack of data on cell type-specific chromatin status of REs. Limitations to publicly available data is likely to be the reason why we could not assign statistical significance to SNPs present in REs. Nevertheless, determining and categorising SNPs may help us understand each patient’s disease progression, individual responses to drugs, or susceptibility to relapse. Some patients had SNPs at the *RUNX1* locus with previously described functional effects on haematopoietic cells. However, the SNP data for myeloproliferative disorders lacks significance because the majority of SNP information comes from analysis of whole blood, rather than each specific cell type. 

## 5. Conclusions

Other than R1RE15 and R1RE21 (which are linked to leukaemia), the relevance of R1RE mutations to the progression of AML is unknown. This study sets the scene for functional analyses to precisely determine how *RUNX1* is regulated, including further *RUNX1* chromatin interaction analyses, and CRISPR/Cas9-mediated interference with RE activity. The results of this informatics analysis also provide a rationale for screening patients with mutations in enhancer regions. By analysing deep sequencing of AML patient samples that have no identifiable mutations in the *RUNX1* coding region or in other leukaemia genes, mutations in REs that lead to dysregulated *RUNX1* expression may be discovered. Sequencing analyses that identify enhancer mutations may not only explain a patient’s AML progression, but could provide the basis for future therapeutic targets. Understanding the regulatory landscape of *RUNX1* will further increase understanding of haematopoiesis as well as identify potential new regions for driver mutations in myeloid malignancies.

## Figures and Tables

**Figure 1 genes-12-01175-f001:**
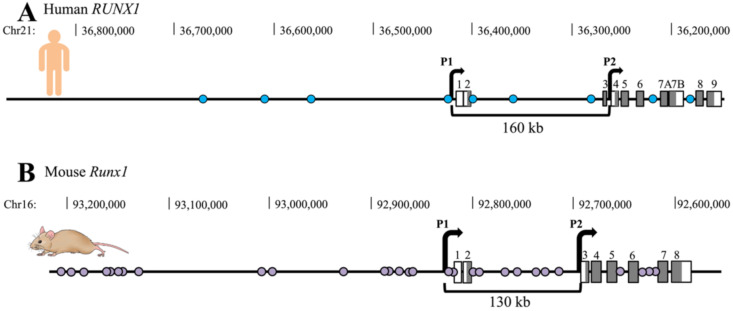
Schematic representing the approximate genomic location of *Runx1* regulatory elements (REs) identified to date in human and mouse. REs are annotated with circles; grey boxes annotate exons of *Runx1*. The two *Runx1* promoters are represented by black right angled arrows (**A**) Genomic location of the 9 previously identified Human *RUNX1* REs. (**B**) Genomic location of the 29 previously identified mouse *Runx1* REs.

**Figure 2 genes-12-01175-f002:**
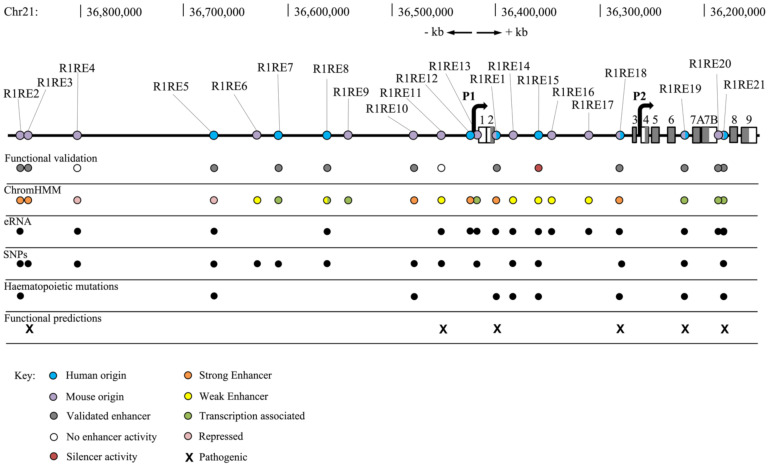
Schematic overview of human *RUNX1* locus and annotations (chromosome 21: 36,148,773-36,872,777). Each RE with human origin is annotated with blue circles, REs with mouse origin are annotated with purple circles, grey boxes annotate exons. The two *RUNX1* promoters are represented by black right angled arrows. R1REs are annotated with their functional validation, ChromHMM characterisation, ability to produce eRNA, SNP status, functional predictions and whether they harbour mutations present in haematopoietic patients. Key for ChromHMM is in the bottom left of the diagram.

**Table 1 genes-12-01175-t001:** Conserved mouse and human *Runx1* regulatory regions.

Name	Location Relative to *Runx1* P1 (kb)	Genome Coordinates (Hg19)	Identification Methods	Interactions Identified	Functional Validation
R1RE1	Mouse and Human *Runx1* relative to P1 +23/+24/eR1/RE1	chr21: 36399106–36399322	Sequence conservation, DNase I hypersensitive sites (DHS) [20,21,23]	Interacts with promoters (P1 and P2) of *Runx1* [23,25].	Haematopoietic enhancer activity in mouse and zebrafish and enhancer activity in 416B, K562, Jurkat; silencer in HEK293 cells [21,23,24,25].
R1RE2	mouse −371	chr21: 36855111–36856085	Interaction with *Runx1* promoters described by 4C and TF binding motif analysis [25].	Interacts with R1RE1 and P1 in HPC7 cells identified by 4C [25].	Hematopoietic enhancer activity in 20–24 hpf zebrafish embryos [25]. Active in pre-haemogenic endothelial cells and intra-arterial clusters in mice [68].
R1RE3	mouse −368	chr21: 36849117–36849289	Sequence conservation and TF binding motif analysis [21].	No available data	Hematopoietic enhancer activity in 20–24 hpf zebrafish embryos [25].
N/A	mouse −354	Not conserved	Interaction with *Runx1* promoters (4C) and TF binding motif analysis [25].	Interacts with R1RE1 and P1 in HPC7 cells [25,27].	Hematopoietic enhancer activity in 20–24 hpf zebrafish embryos [25].
R1RE4	mouse −327 [25]/−328 [24]	chr21: 36800068–36801396	Interaction with *Runx1* promoters (4C) and TF binding motif analysis [25]. Recruits TFs (ERG, FLI1, GATA2, GFI1B, LYL1, MEIS1, SPI1, RUNX1 and TAL1) and H3K27ac [24]	Interacts with R1RE1 and P1 in HPC7 cells [25,27].	No enhancer activity in mice [24] or zebrafish [25].
N/A	mouse −321 [25]/−322 [24]	Not conserved	Interaction with *Runx1* promoters (4C) and TF binding motif analysis [25]. Recruits TFs (ERG, FLI1, GATA2, GFI1B, LYL1, MEIS1, SPI1, RUNX1 and TAL1) and H3K27ac [24]	Interacts with R1RE1 and P1 in HPC7 cells [25,27].	Identified, but not tested for enhancer activity [24]. Hematopoietic enhancer activity in 20-24 hpf zebrafish embryos [25].
N/A	mouse −303	Not conserved	Interaction with *Runx1* promoters (4C) and TF binding motif analysis [25].	Interacts with R1RE1 and P2 in HPC7 cells [25,27].	Hematopoietic enhancer activity in 20-24 hpf zebrafish embryos and in keratinocytes from 20 hpf [25].
R1RE5	human −250/E6	chr21: 36669712–36670621	Recruits EBNA2 [26].	No available data	Enhancer activity in GM12878 and EBV positive Burkitt’s Lymphoma cell lines [26].
R1RE6	mouse −181	chr21: 36629105–36629568	Recruits H3K27Ac and TFs (EOMES, SCL) [33].	No available data	No available data
R1RE7	human −188/E4	chr21: 36608329–36608806	Recruits EBNA2 [26].	No available data	Enhancer activity in GM12878 and EBV positive Burkitt’s Lymphoma cell lines. [26].
N/A	mouse −171	Not conserved	Recruits H3K27Ac and TFs (EOMES, SCL) [33].	No available data	No available data
R1RE8	human −139/E1	chr21: 36561619–36562555	Recruits EBNA2 [26].	No available data	Enhancer activity in GM12878 and EBV positive Burkitt’s Lymphoma cell lines [26].
R1RE9	mouse −101	chr21: 36542872–36543055	Sequence conservation and TF binding motif analysis [21]	No available data	No available data
R1RE10	mouse −58 [25]/−59 [24]	chr21: 36478706–36478906	Interaction with *Runx1* promoters (4C) and TF binding motif analysis [25]. Recruits TFs (ERG, FLI1, GATA2, GFI1B, LYL1, MEIS1, SPI1, RUNX1 and TAL1) and H3K27ac [24]	Interacts with R1RE1 and P1 in HPC7 cells [25].	Haematopoietic enhancer activity in E11.5 transgenic mice and enhancer activity in 416B cells [24]. Hematopoietic enhancer activity in 20–24 hpf zebrafish embryos [25]
N/A	mouse −48	Not conserved	Interaction with *Runx1* promoters (4C) and TF binding motif analysis [25].	Interacts with R1RE1 and P1 in HPC7 cells [25].	Hematopoietic enhancer activity in 20–24 hpf zebrafish embryos [25].
R1RE11	mouse −43	chr21: 36464084–36464260	Recruits TFs (ERG, FLI1, GATA2, GFI1B, LYL1, MEIS1, SPI1, RUNX1 and TAL1) and H3K27ac [24]	No available data	No enhancer activity in mice [24].
N/A	mouse −42	Not conserved	Recruits TFs (ERG, FLI1, GATA2, GFI1B, LYL1, MEIS1, SPI1, RUNX1 and TAL1) and H3K27ac [24]	No available data	No enhancer activity in mice [24].
R1RE12	human −5	chr21: 36423511–36423652	SON binding in MEG-01 and CMY cells [29]	No available data	No available data
N/A	mouse +1	Not conserved	Recruits TFs (ERG, FLI1, GATA2, GFI1B, LYL1, MEIS1, SPI1, RUNX1 and TAL1) and H3K27ac [24]	No available data	No available data
R1RE13	mouse +3	chr21: 36418472–36418744	Recruits TFs (ERG, FLI1, GATA2, GFI1B, LYL1, MEIS1, SPI1, RUNX1 and TAL1) and H3K27ac [24]	No available data	Haematopoietic enhancer activity in E11.5 transgenic mice and enhancer activity in 416B cells [24].
R1RE14	mouse +32	chr21: 36384185–36384451	Sequence conservation and TF binding motif analysis [21]	No available data	No available data
R1RE15	human +62	chr21: 36358933–36359953	ChIA-PET (RNA polymerase II) and DHS; conserved sites for GFI1/GFI1B and SNAI1.	Interacts with P2 in K562 and OCl-AML3 cell lines [28]	Silencer activity in K562, OCl-AML3, U937 cell lines [28].
R1RE16	mouse +59	chr21: 36346597–36346844	Sequence conservation and TF binding motif analysis [21]	No available data	No available data
N/A	mouse +64	Not conserved	Sequence conservation and TF binding motif analysis [21]	No available data	No available data
R1RE17	mouse +87	chr21: 36310873–36311013	Sequence conservation and TF binding motif analysis [21]	No available data	No available data
N/A	mouse +99	Not conserved	Sequence conservation and TF binding motif analysis [21]	No available data	No available data
R1RE18	mouse +110 [21]/human +139 [29]	chr21: 36280710–36281200	Sequence conservation and TF binding motif analysis [21]. SON binding in MEG-01, CMY, and CMK cells; increased H3K4 methylation upon SON depletion [29]	Interacts with P1 in HPC7 cells identified by 4C [25].	Haematopoietic enhancer activity in E11.5 transgenic mice and enhancer activity in 416B cells [24]. Hematopoietic enhancer activity in 20–24 hpf zebrafish embryos [25].
R1RE19	mouse +171 [32]/human +43kb [30]	chr21: 36218040–36218420	DHS specific to AML samples (FLT3-ITD) [30]. GATA3 binding [32]	No available data	Enhancer activity in UG26-1B6 cells [32]
N/A	mouse +181	Not conserved	Recruits TFs (ERG, FLI1, GATA2, GFI1B, LYL1, MEIS1, SPI1, RUNX1 and TAL1) and H3K27ac [24]	No available data	No enhancer activity in mice [24].
R1RE20	mouse +199	chr21: 36186002–36186045	Sequence conservation, p63 binding [31]	No available data	Enhancer activity in PTK2 cells [31]
R1RE21	human intron 5.2 [23] containing mouse +204 [24]	chr21: 36179311–36181581	Sequence conservation; DHS, [23] Recruits TFs (ERG, FLI1, GATA2, GFI1B, LYL1, MEIS1, SPI1, RUNX1 and TAL1) and H3K27ac [24]	Interacts with P1 and P2 in K562, Jurkat, HEK293 cell lines [23]	No enhancer or silencer activity observed in K562, Jurkat and HEK293 cell lines [23]. Haematopoietic enhancer activity in E11.5 transgenic mice and enhancer activity in 416B cells [24].

**Table 2 genes-12-01175-t002:** Annotations of identified R1REs in K562 cells.

Name	Location Relative to *Runx1* P1 (kb)	ENCODE Annotation	ChromHMM Annotation	Altered ATAC Accessibility in Cohesin *STAG2*-/-
R1RE1	mouse and human *Runx1* relative to P1 +23/+24/eR1/RE1	DHS, RNA Pol II, SMC3, CTCF, p300, LSD1, H3K27ac, H3K4me1, H3K4me3, H3K9ac, H3K9me3, GATA1, SPI1, TAL1, GATA2	Strong Enhancer	No change
R1RE2	mouse −371	DHS, RNA Pol II, CTCF, p300, LSD1, H3K27ac, H3K4me1, H3K4me3, H3K9ac, H3K9me3, H3K27me3, GATA1, TAL1, GATA2	Strong Enhancer	Increased
R1RE3	mouse −368	RNA Pol II, p300, LSD1, H3K27ac, H3K4me1, H3K4me3, H3K9ac, H3K9me3, H3K27me3	Strong Enhancer	No change
R1RE4	mouse −327 [25]/−328 [24]	p300, H3K4me1, H3K9ac, H3K9me3, H3K27me3	Repressed	No change
R1RE5	human −250/E6	RNA Pol II, p300, LSD1, H3K9me3, H3K27me3	Repressed	No change
R1RE6	mouse −181	DHS, RNA Pol II, p300, LSD1,H3K27ac, H3K4me1, H3K4me3, H3K9ac, H3K9me3	Weak Enhancer	Increased
R1RE7	human −188/E4	p300, H3K4me3, H3K9me3, H3K27me3, SPI1	Transcription Associated	No change
R1RE8	human −139/E1	p300, H3K4me1, H3K4me3, H3K9ac, H3K9me3, SPI1	Weak enhancer, Transcription Associated	No change
R1RE9	mouse −101	RNA Pol II, p300, LSD1, H3K27ac, H3K4me1, H3K4me3, H3K9me3,	Transcription Associated	No change
R1RE10	mouse −58 [25]/−59 [24]	DHS, RNA Pol II, CTCF, p300, LSD1, H3K27ac, H3K4me1, H3K4me3, H3K9ac, H3K9me3, GATA1, TAL1, GATA2	Strong Enhancer	No change
R1RE11	mouse −43	RNA Pol II, CTCF, p300, LSD1, H3K4me3, H3K9ac, H3K9me3	Weak Enhancer	No change
R1RE12	human −5	DHS, RNA Pol II, RAD21, CTCF, p300, LSD1, H3K27ac, H3K4me1, H3K4me3, H3K9ac, H3K9me3	Strong Enhancer	No change
R1RE13	mouse +3	RNA Pol II, CTCF, p300, LSD1, H3K4me1, H3K4me3, H3K9ac, H3K9me3	Transcription Associated	No change
R1RE14	mouse +32	DHS, RNA Pol II, p300, LSD1,H3K4me1, H3K4me3, H3K9ac, H3K9me3	Weak Enhancer	No change
R1RE15	human +62	DHS, RNA Pol II, CTCF, p300, LSD1, H3K4me1, H3K4me3, H3K9ac, H3K9me3, H3K27me3	Weak Enhancer	No change
R1RE16	mouse +59	RNA Pol II, CTCF, p300, LSD1,H3K4me1, H3K4me3, H3K9ac, H3K9me3	Weak Enhancer	No change
R1RE17	mouse +87	RNA Pol II, CTCF, p300, LSD1, H3K4me1, H3K4me3, H3K9me3	Weak Enhancer	No change
R1RE18	mouse +110 [21]/human +139 [29]	DHS, RNA Pol II, RAD21, p300, LSD1, H3K27ac, H3K4me1, H3K4me3, H3K9ac, H3K9me3, H3K27me3 GATA1, TAL1, GATA2	Strong Enhancer	Increased
R1RE19	mouse +171 [32]/human +143 [30]	DHS, RNA Pol II, CTCF, p300, LSD1, H3K27ac, H3K4me1, H3K9ac, SPI1, TAL1	Transcription Associated	No change
R1RE20	mouse +199	RNA Pol II, CTCF, p300, LSD1, H3K9me3	Transcription Associated	No change
R1RE21	human intron 5.2 [23] containing mouse +204 [24]	DHS, RNA Pol II, CTCF, p300, LSD1, H3K27ac, H3K4me1H3K9ac, H3K9me3, H3K27me3	Transcription Associated	Increased

## Data Availability

Publicly available datasets were analysed in this study. These data can be found here: https://genome.ucsc.edu/ (accessed on 30 June 2021), https://www.ebi.ac.uk/gwas/, https://cancer.sanger.ac.uk/cosmic, https://dice-database.org/, https://pubs.broadinstitute.org/mammals/haploreg/haploreg.php, https://gtexportal.org/home/, https://gnomad.broadinstitute.org/, https://dcc.icgc.org/, https://www.cbioportal.org/, https://www.cancer.gov/about-nci/organization/ccg/research/structural-genomics/tcga, http://bioinfo.au.tsinghua.edu.cn/dbsuper,
https://doi.org/10.1093/jmcb/mjz114, https://doi.org/10.1038/s41588-020-0578-5.

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
