# Peer review of "Transcriptional Regulation of RUNX1: An Informatics Analysis"

_genes, 2021, doi:10.3390/genes12081175_

Round 1
Reviewer 1 Report
In the paper “Transcriptional regulation of RUNX1: a meta-analysis”, Amarni and colleagues characterized the regulatory landscape of vertebrate Runx1. They identified gene regulatory elements conserved between mouse and human and they added information retrieved in several publicly available datasets (ENCODE, UCSC, GWAS catalog, HaploReg, GTEx, cBioportal, TCGA) associated to these regions.
The paper is well written and the explanation is clear. The first part is interesting, but the second one is too general and too speculative. This paper could be a good start to perform a more MDS-related work with clinical data or with experimental validation in human cells.
The paper is described as a meta-analysis. A meta-analysis is a statistical analysis that combines the results of multiple scientific studies. This work lacks of statistical tests, because the amount of data associated to RUNX1 promoter was low. Moreover, they use one dataset for each analysis. In this way, the authors can’t aggregate data from different approaches in order to identify new helpful information that original papers did not find.
These are my comments:
Major comments
- In the paper you found some SNPs that fall in REs regions. Did you perform any statistical test to affirm that these intersections didn’t happen by chance?
- The paper talks about the possible involvement of RUNX1 REs in haematological malignancies development. In these syndromes RUNX1 it is usually mutated in the exonic regions or it is involved in chromosomal aberrations. Using TCGA or similar dataset, did you check the percentage of MDS/AML patients with RUNX1 expression deregulation without concurrent DNA aberrations? How many patients could be influenced by changes in RUNX1 REs?
- In order to make a more MDS/AML related analysis, why you did not use COSMIC non-coding regions mutations? Or why you did not search for other MDS/AML related datasets?
- Did you check for any data obtained by other MDS/AML cell lines?
- To understand the powerful of the data obtained by your analysis, you must write the number of patients with myelodysplastic syndromes deposited in each database.
Author Response
Thank you for pointing out that the term ‘meta-analysis’ is not accurately applied. We have renamed the paper to “Transcriptional regulation of RUNX1: an informatics analysis”
- Reviewer comment: In the paper you found some SNPs that fall in REs regions. Did you perform any statistical test to affirm that these intersections didn’t happen by chance?
Thank you for the suggestion to reinforce the statistical significance of RE SNPs. We had already investigated the likelihood of a SNP altering RUNX1 expression in haematopoietic cells by using CoDeS3D in the original draft of the paper. In the revised manuscript, we additionally use FATHMM (Functional Analysis through Hidden Markov Models), a statistical test that predicts the likelihood of an identified mutation to have a pathogenic effect.
CoDeS3D: We used the CoDeS3D pipeline, which statistically interrogates whether SNPs spatially associate with genes that also have differential expression depending on the SNP allele (Expression Quantitative Trait loci). Despite finding many regulatory spatial interactions between RUNX1 regulatory region SNPs and RUNX1, none of them were supported by a significant eQTL interaction (FDR< 0.05). Our conclusion is that none of the SNPs were significant in CODES3D because eQTL data in GTEX was not sufficiently powered to generate statistics. The GTEx portal sampled the 54 tissues analysed for eQTLs from healthy older individuals. GTEx also only measures whole blood, rather than individual cell types, so cell type- or disease-specific effects may not be detected. We predict there is an underestimation of SNP significance because many MDS/AML datasets are exome and exclude the regulatory regions. We already discuss the limitations of these data, but have added the following sentences to the discussion “The present study is limited by both the paucity of genomic sequence data of REs in normal and leukaemic cells, and the lack of data on cell type-specific chromatin status of REs. Limitations to publicly available data is likely to be the reason why we could not assign statistical significance to SNPs present in REs.”
FATHMM: FATHMM non-coding predictor uses available ENCODE data (conservation, histone modification, TFBS, and open chromatin) to analyse the likelihood of a mutation to be pathogenic. Predictions with values above 0.5 are predicted to be pathogenic/deleterious, while those below 0.5 are predicted to be neutral or benign [1,2]. Some of the mutations are predicted to be pathogenic by FATHMM (R1RE1 ICGC mutation, R1RE3 rs116951441, R1RE11 rs2834756, R1RE18 rs73900579, R1RE19 rs2284613 and the previously AML susceptibility associated SNPs R1RE21 rs2249650 and rs2268276). We have added the FATHMM predictions to the manuscript (New Supplementary Table 3, Updated Figure 2, Methods 2.5, Results 3.6, Discussion lines 364-365,372-379 ). These data are limited in prediction ability because FATHMM calculates pathogenicity based on available ENCODE data for the sequence. FATHMM cannot predict cell type-specific pathogenicity, or the impact of deletion or insertion mutations.
- Reviewer comment: The paper talks about the possible involvement of RUNX1 REs in haematological malignancies development. In these syndromes RUNX1 it is usually mutated in the exonic regions or it is involved in chromosomal aberrations. Using TCGA or similar dataset, did you check the percentage of MDS/AML patients with RUNX1 expression deregulation without concurrent DNA aberrations? How many patients could be influenced by changes in RUNX1 REs?
The reviewer makes a good point. We analysed 451 AML samples from cBioportal to determine the level of expression of RUNX1 according to mutation status, and shows that expression of RUNX1 in AML is highly variable regardless of RUNX1 mutation status, making it plausible that altered RUNX1 in patients with no mutation could potentially be due to RE mutations [3,4]. We have added this information to the paper (New Supplementary Figure 1, lines 298-300).
- Reviewer comment: In order to make a more MDS/AML related analysis, why you did not use COSMIC non-coding regions mutations? Or why you did not search for other MDS/AML related datasets?
DICE (GRCh37/hg19) was also investigated however did not report any SNPs in the R1REs [5,6].No SNPs are found in the REs in the COSMIC database (GRCh37/hg19 and GRCh38/hg38) [7].
However, COSMIC reports mutations in haematopoietic and lymphoid cells in R1RE15, R1RE18 and R1RE21. None of these mutations are predicted to be pathogenic by the FATHMM prediction scores.All mutations are predicted by FATHMM to be neutral, R1RE15 mutation has a predicted score of 0.09, R1RE18 has a predicted score of 0.21 and R1RE21 has a predicted score of 0.01.
These datasets are included in the revised manuscript (New Supplementary Table 3, Lines 109-110, 131-132,287-2289).
- Reviewer comment: Did you check for any data obtained by other MDS/AML cell lines?
Limited data from other cell lines are certainly available, but differences in the depth of the data makes comparisons between cell lines at the RUNX1 regulatory elements difficult. The reason we used K562 is because it is among the most comprehensively investigated (tier 1) cell lines in ENCODE. Comprehensive annotation of K562 by ENCODE allowed us to have some confidence in assigning regulatory activity to the REs described in our study. In the future, other myeloid cell lines and primary cells will be valuable once they have been analysed to the same depth as K562, but unfortunately there aren’t any other lines as informative as K562 to date. The limitations of our data are addressed in the discussion.
- Reviewer comment: To understand the powerful of the data obtained by your analysis, you must write the number of patients with myelodysplastic syndromes deposited in each database.
Thank you for pointing out this omission. The numbers are as follows: 1,732 Whole genome sequenced (WGS) AML/MDS samples from ICGC database, 50 WGS AML cases in TCGA dataset, COSMIC had an 39 additional WGS AML samples. 451 AML samples from cBioportal of which 50 are WGS. These numbers are now listed in the methods.
Reviewer 2 Report
This manuscript is a nice study by Thomas et al, where the authors attempted to meta-analyze the RUNX1 locus. The simple study design, yet an impactful outcome, is appreciated and I believe this would be
important for the community. However, I have a comment at this stage which is highlighted below. If the other reviews are favorable, I would like the authors to discuss this in the discussion section before this can be published.
- besides analyzing the K562 cells, were data from primary human erythroid cells interrogated at all? How does that compare/contrast/add additional value to the conclusions?
Author Response
- Reviewer comment: besides analyzing the K562 cells, were data from primary human erythroid cells interrogated at all? How does that compare/contrast/add additional value to the conclusions?
Thank you for the suggestion. While chromatin data for primary lines is available for analysis, no dataset has all the chromatin marks of interest in combination. The limitations of the data are addressed in the discussion.
Round 2
Reviewer 1 Report
The authors replied all my questions.
Please cancel "meta-analysis" word in the text, in particular in conclusions section